# The Multilevel Pathway in MSTs for the Evaluation and Treatment of Parents and Minor Victims of ACEs: Qualitative Analysis of the Intervention Protocol

**DOI:** 10.3390/children9030358

**Published:** 2022-03-04

**Authors:** Luna Carpinelli, Daniela D’Elia, Giulia Savarese

**Affiliations:** Department of Medicine and Surgery, University of Salerno, 84081 Baronissi, Italy; lcarpinelli@unisa.it (L.C.); ddelia@unisa.it (D.D.)

**Keywords:** adverse childhood experiences (ACEs), parenting, minors, treatment, intervention

## Abstract

Background: Adverse childhood experiences (ACEs) may be an important risk factor for the onset of developmental psychopathological disorders. Families involved in ACEs are often the subject of social or welfare policies aimed solely at the victim, without a proper consideration of family functionality. Methods: We describe the results of an Italian intervention project, which aimed to reinforce both the local networking of the Campania region, and the skills of the operators involved in actions to prevent and combat ACEs. The project was characterized by different phases and two actions, namely: (1) diagnosis and therapy aimed at child victims of ACEs and their families; (2) supervision of the operators of the multidisciplinary specialized teams (MSTs). Results: 99% of the cases under review were characterized by intra-family violence; 34% suffered psychological abuse, 33% neglect, 23% inappropriate care, 4% sexual abuse, 3% excessive care and 3% physical abuse. Conclusions: Thanks to the interventions carried out, severe and chronic ACE situations were recognized, and processes of de-institutionalization and the construction of life projects were carried out in accordance with the territorial services. This offered child victims and their families an opportunity to restore the conditions of well-being, both for the growth of the individual and the family system.

## 1. Introduction

In recent years, the operators of local social and health services have found limitations in the practices and techniques they use in dealing with forms of hardship within families and more complex difficulties. The discomfort of, and in, families is more often connected to the affective dimensions, the internal relational context that can be influenced by structural configurations, but which is probably conditioned above all by the stories and subjective characteristics of the individuals, by the demands for claims that are harbored for years, from the relationships that bind, even in a pathological way, and from the pushes to subjugate the other in order to affirm one’s superiority.

In its Action Plan 2013–2020 [1], the World Health Organization states: “Exposure to stressful events at a young age is an established risk factor for the onset of mental disorders that can be preventable”. The so-called “adverse childhood experiences” (ACEs) (i.e., negative life events, trauma, bereavement) would in fact be considered non-specific factors [2] capable of increasing the probability of the appearance of any physical and mental illness, of influencing the course of such an illness, worsening the prognosis, or causing relapses in the case of basically chronic pathologies. The following experiences lived within the family context before the age of 18 [3] can be classified as ACEs: recurrent physical and/or psychological abuse; sexual abuse; the presence in the family unit of a person dependent on alcohol or substances; the presence within the family of a person charged with a crime; a family member with overt mental disorders, institutionalized or suicidal; witnessing violence; the presence of only one or no parent; and physical and/or emotional neglect.

In recent years, more attention and research has been focused not only on abuse and mistreatment, but also on physical and emotional neglect and exposure to domestic violence. Some studies in past decades have shown that children living in a climate of violence had higher levels of anxiety or a greater frequency of post-traumatic stress disorder [4] and that, in the majority of cases, the ACEs were not isolated and, when present, there tended to be more than one such experience. The hypothesis that ACEs during development may constitute an important risk factor for the onset of psychopathological disorders both during development and in adulthood seems to have several data sets to support it. That this risk factor has its roots not only in more purely psychological and cognitive aspects, but also in neurobiological phenomena favoring emotional dysregulation appears to be a hypothesis supported by current knowledge on the physiology of the brain. That emotional dysregulation of this type can constitute, in some situations, a significant risk factor (in correlation with neuroendocrine and neuroimmunological alterations) for the onset of somatic pathologies is currently a highly suggestive hypothesis, at the basis of which there are now significant clues. Importantly, such evidence is supported by epidemiological investigations, not only by clinical reports.

Families that have individuals in difficulty are often the subject of social or welfare policies aimed solely at the problematic subject, without taking into consideration the family as a whole. Sometimes, “multiproblematic” or “dysfunctional” families are cared for by different services, between which there is no connection around the family as a whole. Specialized services initially require the motivation of people. Some deal with the problem of “pathology”, of the “dependence” of their users, so it happens that the work related to the parenting sphere and protecting and helping children, in a rationale of still fragmented, sectorial, not yet recomposed competences, is in fact the burden of basic social services. Thus, an overall and shared assessment of situations, even highly problematic ones, which may require protection interventions, including judicial ones, is lacking. Different needs correspond to different services, but the family is always the same; the family must always feel integrated (i.e., a global being).

## 2. Intervention and Prevention Program

In this study, we will highlight the results of the *“Program of interventions aimed at the prevention of abuse and maltreatment of minors”* project funded by the Campania region (Italy). This project was carried out between 2016 and 2018 by the University of Salerno in partnership with the EOS Onlus Foundation of Naples (at the time “G. Toniolo” Consultorio) service combating abuse and maltreatment, in which a series of specialized awareness-raising, information and training interventions were carried out to reinforce both Campania territorial networking and the skills of the operators involved in actions to prevent and combat child abuse and maltreatment. The project was characterized by two phases: “Action 1” relating to diagnosis and therapy aimed at child victims of ACEs and their families; and “Action 2” dedicated to the supervision of the operators making up the multidisciplinary specialized teams (MST).

Specifically, the intervention relating to the establishment of the MSTs, who take care of families and minors at risk [5], was based on the “ecological model” of [6,7], the fundamental principle of which is that no factor in its own right can fully explain why individuals act violently towards other people, or how such violent attitudes are prevalent in some societies but not others. Furthermore, the “ecological model” constitutes a valid all-round indication as it considers four concentric interacting areas in the etiology and repair of damage, consisting of: the ontogenetic level or individual factors; the microsystem level or family factors; the exosystem level or the social and economic factors relating to the community to which it belongs; and the macrosystem level or the institutional and cultural determinants [6]. From this derives an intervention model that conceives prevention and protection in an integrated way involving the different levels of life, providing a global and articulated basis of options for detection and for “treatment” interventions [7].

In the traumatological dimension, the “cure” can only be articulated around two synergistic pillars; that is, psychotherapy and the promotion of a reparative, factual and emotional experience, which have the dual task of destabilizing what is fixed in the post-operative models. Leading the subject to reattempt to attain positive attachment experiences can be traumatic. In other words, there is a need to establish an intervention aimed at changing the system of the meaning of victim (and sometimes also of those who work with her); that is, the complex thoughts and feelings that constitute our “philosophy” regarding the functioning of the world. This must be combined with a “human” approach, focusing on the victim’s concrete experience, that constructs another world in which it is really possible to feel safe from everything that one has escaped from, and which is based on laws and constants that make it possible to change the system of previously learned meanings. Since the trauma occurred in the real world, the pain of looking at it, trying to understand it, tolerating it and processing it will never be bearable or appear sensible and advantageous if there is no guarantee that a concrete life alternative really exists.

In this sense, protection and reparation are connected in the reconstruction of meaningful and reassuring ties with family members, friends and informal relationships, which represent significant experiences that can support a victim in living an autonomous life and attaining emotional well-being, both for the minor and for the parents.

In order to pursue such ambitious goals, the moment of taking charge begins—as well as the social anamnesis—with the psychodiagnostic evaluation of post-traumatic markers, both for parents and for their children, as the process of assessment is understood as being the first substantial step in therapy [8]. All this allows important actions that promote family strength and well-being in children, allowing us to enhance the “protective factors” and resources of that child and his familiar system.

## 3. Methodology

### 3.1. Action 1: Level II Assessment and Diagnostic Consultancy

The psychodiagnostic evaluation of minors is based on the specialist evaluation model for post-traumatic functioning, proposed by Marinella Malacrea [7], which includes three levels of in-depth analysis through the use of defined tools with low, medium and high diagnostic impact. Each level is then divided into two observation axes: the axis of manifest behavior, and the axis of internal experiences. Specifically, the low-impact tools investigate the cognitive and adaptive abilities, resources and adaptations of the subject. These tools are the Child Behavior Checklist (CBCL) [9] and the genogram for behaviors; while, for the internal world, free drawing and the Story Stem Battery (SSB) are used [10].

The medium-impact tools access the channels of projective fantasies in a less controllable way and are aimed at analyzing behavior. They are: the Trauma Symptom Checklist for Children (TSCC) [11]; the Parental Stress Index (PSI-4) [12]; and the Trauma Symptom Checklist for Young Children (TSCYC) [13]; and for the internal world, the Family Attitudes Test (FAT); [14] a symbolic game (also in light of the indicators of the post-traumatic game described by Terr in 1991); themed drawings; the Children’s Apperception Test (CAT) [15]; and the Thematic Apperception Test (TAT) [16].

The high-impact tools tend to bring out the deepest experiences, removed from conscious control. The Rorschach test [17] is used and a reading specifically aimed at capturing the indicators of post-traumatic functioning is adopted [18]. On the behavioral level, there are tests aimed at investigating pathological behaviors already highlighted by previous tests, with a focus on specific post-traumatic reactions, such as sexualized behaviors with the Child Sexual Behavior Inventory (CSBI) [19], depressive traits with the Children’s Depression Inventory (CDI-2) [20], and dissociative traits with the Dissociative Experiences Scale (DES) [21] for children and adolescents.

Furthermore, in recent years, together with clinical work in the therapeutic field through the elective approach to the treatment of trauma EMDR (acronym for Eye Movement Desensitization and Reprocessing; Shapiro [22]), this reference test battery is also being integrated with more specific tests for evaluating traumatic attachment, such as the Manchester Child Attachment Story Task (MCAST) [23].

Regarding the evaluation of parental recoverability, it is equally configured as a priority clinical intervention and takes into account the CISMAI *Guidelines for clinical evaluation and activation of the recovery of parenthood in the psychosocial path of protection of minors,* integrating tools and formats of specific and diversified intervention [24]; a multi-modal, multi-model, integrated multi-orientation approach that aims to activate possible resources and potential within dysfunctional parenting dynamics for the purpose of their possible recovery.

The proposed evaluation and treatment model is based on the theoretical hypothesis that the abusive parent is in turn a child who is still injured and traumatized, and therefore also has a history of early traumatization behind them, in line with Fellitti’s investigations [25] in the field of ACEs. The evaluation process lasts six months and is carried out both through joint and individual clinical couple interviews, and through a specific battery of diagnostic reagents configured ad hoc, which include: the Human Figure Test (Machover, 1980), the Rorschach test [17], the MMPI-2-R [26] and the Parental Stress Index (PSI-4) [12] (already considered in the evaluation of the child).

This intervention has as its final objective the formulation of a prognosis regarding the possibility of recovering adequate parenting, taking the form of an in-depth work on the parental skills present at the time of taking charge (which in most cases, as is evident, is compromised), and then treatment of the same in order to experiment with the transformative possibilities in it within the clinical path.

The actions envisaged in “Action 1” are carried out by specialized professional figures, i.e., by psychologists and psychotherapists with specific training in diagnostic evaluation. The interviews conducted by these professionals are structured on the protocol described, using the interviews already defined by each test administered to both minors and parents.

### 3.2. Action 2: Supervision of MSTs

The clinical supervision of MSTs, made up of psychologists, social workers, educators and child neuropsychiatrists, has a double relevance: on one hand, the unthinkable nature of the issues addressed relating to the detection of complex trauma supports the risk of prolonging the time of diagnosis; and on the other hand, the burdensome intertwining with judicial processes exposes the clinician to often having to support the outcome of the path (of the psychodiagnostic evaluation of the child and recoverability of the parent) with a heavy experience of responsibility towards the child and towards the possibility of seeing the ties with parents cut off. Supervision from this point of view reinforces the experience of professional competence and efficacy, making it possible for the clinician to expose himself, through his own professional judgment, to the judicial context.

The supervision meetings on child care focused on psychodiagnosis as an analysis of the prevailing post-traumatic functioning, not only in terms of descriptive diagnosis, and, compared to the clinical intervention, much space was also dedicated to EMDR supervision.

The supervision of the evaluation of parental recoverability worked on the possibility of comparing it with the experimentation of a clinical intervention model that conceptualizes the evaluation as a procedural diagnosis, and not referring to it as a nosographic label.

### 3.3. Clinical Treatment

The phase of treatment and care of abused children and dysfunctional parenting involves the use of EMDR as an elective therapeutic approach. EMDR is a valuable resource in clinical intervention that focuses, in the case of children, on the processing of life-threatening traumas and attachment traumas, as well as on strengthening resources. In the case of parents, the treatment focuses on the management of the experience of care, on the experiences of the parent associated with it and on a profound and guided reflection on their own mental states, along with the mental state of the child in the interaction [27,28,29,30].

We know, in fact, that traumatic experiences in the parent, stored in a dysfunctional way, can reactivate in the parent’s care system. The use of EMDR in the area of parenting is therefore particularly effective for the elaboration and analysis of implicit models of dysfunctional relationship in the parent–child relationship, helping to access the traumatic memories inherent in one’s own attachment history and process them with an adaptive resolution. The EMDR approach can, therefore, be used to help break the intergenerational transmission of traumatic and dysfunctional care [31].

As the patient identifies and processes his early traumas with the help of EMDR, he gradually becomes able to separate the present from his past, narrating a coherent autobiography.

The phases envisaged in “Action 2” were conducted by psychologists and psychotherapists trained in EMDR. This therapeutic approach uses a specific and detailed clinical protocol, in which the therapist guides the assessment subject of EMDR.

During the clinical interviews, the patient’s personal history and the reported problem are reconstructed, allowing the therapist to identify the target memories on which to work, with the construction of an adequate therapeutic plan. It is always a crucial phase as, in addition to laying the foundations for building a relationship of trust with the therapist, it allows us to focus more clearly on some aspects of ourselves.

## 4. Results

### 4.1. Characteristic Analysis of the Users Taken in Charge

The emergence work was carried out from January to July 2018 and involved 50 professionals: 32 social workers, 8 community coordinators and 10 psychologists.

A total of 28 meetings were conducted, in which 15 were taken in clinical charge for a total of 35 subjects (39% minors and 61% parents), divided into the 5 provinces of the Campania region.

Specifically, 67% of the minor victims of ACEs were males and 33% females, with a strong prevalence of preteens and adolescents (aged 11–18 years), to the detriment of children (4–10 years).

In total, 82% of the situations taken in clinical charge were in the midst of a judicial protection process, while 18% took action spontaneously at the request of the care or service employers, but without the prescription of the Judicial Authorities.

Regarding the type of ACE, we note that 99% were characterized by intra-family violence; specifically, 34% suffered psychological abuse, 33% neglect, 23% inappropriate care, 4% sexual abuse, 3% excessive care and 3% physical abuse.

### 4.2. Evaluation of the Results of the Interventions

With respect to the type of interventions, it is noted that the integrated management of the four regional MSTs made it possible to define the post-traumatic outcomes on the psychophysical health of minors (50% of the sample) who were institutionalized for long periods and not treated for their symptoms (34% psychodiagnosis and support, plus 16% psychodiagnosis). Furthermore, parental recoverability assessment interventions were carried out in 34% of cases, of which, however, only 3% of the sample was able to access a positive prognosis of treatability and take advantage of a therapeutic space for their frailties.

At each meeting, on a monthly basis, the operators of the different teams brought a clinical case through clinical interviews and psychological tests administered for the evaluation phase, and the EMDR protocol for the treatment phase.

Step I: The first phase of the supervision process involved an initial presentation of all the members of the group, centered on a description of the individual professions and the various services to which they belonged, which allowed us to explore and promote connections between operators and work contexts, often destined to remain “isolated monads” crushed by daily operations. This also made it possible to establish new relationships that lead to constructive collaborations. An attempt was made to establish a working climate based on trust and mutual collaboration, within which each participant could expose themselves in an authentic way, recognizing their own internal resonances and also sharing any “mistakes”, as well as professional successes. In this first phase of the supervision work, the intervention process model was also presented in order to standardize the intervention practices and procedures in the various professional teams, detailing spaces, times and protagonists of the clinical consultation setting for the evaluation of parental recoverability.

Step II: This phase of the supervision work focused on the start of the clinical work of evaluating parental recoverability, focusing attention on the knowledge and definition of the institutional framework in which the intervention takes place (mandate, decree, sending), and on the re-construction of the history of the family unit as an indispensable premise for clinical management. In particular, these defensive structures were explored as possible reactions of the parent in achieving the first phase of knowledge and the sharing of the reasons for the current evaluation of parental skills. (1) Denial of the facts: the parent proclaims his innocence and extraneousness, accusing the victim of lying (extreme level). (2) Denial of awareness: there is consensus on the reality of the harm suffered by the child, but his own explanation is given for the harm (effect of alcohol, drugs or semi-sleep). (3) Denial of responsibility: both the facts and the awareness of having committed them are admitted, but one’s responsibility is not admitted, and is instead usually attributed to others (i.e., the victim, the other parent, extended family, etc.) (4) Denial of impact: full or partial acknowledgment of facts, awareness and responsibility, and minimization of the consequences of one’s conduct on the child.

The level of parental denial and the severity of the damage inflicted on minors constitute the two elective criteria for deciding whether to report the case to the Juvenile Court, thus activating a context of forced intervention, or whether to lean towards a spontaneous context, where there is a good level of recognition of the maltreatment by part of the family such that they decide to invite the parents to a clinic, where they will be able to undertake a path of recovery. It must be emphasized that, between the two criteria (levels of denial/awareness and severity of maltreatment), in the choice of whether to report or the less “weighs” more the second, since it is necessary in the case of a serious violence report, even if the parents are aware of the damage they have caused.

Step III: The supervision work in this phase focused in particular on the analysis of the clinical material that emerged for the purpose of defining the attachment style, and for the caring effect of each parent. This also aims to build the premise for something that would help the parent make sense of his or her own experience as a child through a reconstruction of their childhood history within an empathic and containing relationship, which acts as a safe space. Only within such a space is the parent capable of facing the feelings of abandonment, neglect and rejection that they themselves experience, and deal with unprocessed and unexpressed experiences for the first time. To this end, in this phase, much work has been done to reach an attribution of meaning and a re-reading of the behaviors of abusive parents through a shared hypothesis, structured according to the traumatic experiences lived in their evolutionary course, in order to activate and consolidate the possibility of an autonomous motivation for evaluation.

Step IV: In this phase, the supervision work focused on setting up the clinical diagnostic work for the personological assessment through the structuring of joint, individual couple interviews and the administration of the diagnostic reagents required by the assessment protocol. Clinical attention was focused on the analysis of some criteria considered fundamental for the evaluation of parenthood: ego functioning and capacity/inability to adhere to reality; capacity/inability to control impulses; capacity/inability of emotional-affective regulation, capacity/inability to build an object relationship; empathic resources; and maturation of processing, reflection and mentalization skills. We worked for long periods on the possibility of building links, together with the parent, between his damaged parts and his prejudicial behaviors, in order to start working on the possibility that he fully recognizes the suffering that has been generated in his child as a consequence of his dysfunctional behaviors.

Step V: In this phase of the supervision work, different formats were proposed to the groups. We aimed to encourage the parents to read the child’s psychological needs and understand when care of the child was substantially unfulfilled. We also aspired to promote an understanding and mentalization of the post-traumatic functioning of their children, and of the specific damages that emerged from the psychodiagnostic evaluation that was carried out, occurring as a result of their own prejudicial behaviors. In this phase, the supervision groups go through a difficult phase in which the “caregivers” of the children, identifying the pain and trauma of the young victims, push to negatively label the parents and caregivers of the parents who risk “idealizing” the changes, albeit minimal, recorded in the consulting room, without considering the real repercussions on the effective restorative actions implemented in the exercise of their parenting.

Step VI: In this phase, we worked with the supervision groups to devise an articulated and complex definition of the final prognosis regarding parental recoverability. This constituted clinical work that involved the integration of different material coming from both clinical interviews and textological material. These were integrated with information from the different operators involved in answering questions about the prognostic indicators of therapeutic treatability, and the cooperative support of these groups that work towards a common project. The following were considered to be indicators of the recovery of parental competence with a restorative value in the CISMAI guidelines [32,33]: (a) a reduction in the defensive mechanisms of denial; (b) an understanding of, and co-participation in, the child’s suffering; (c) the ability to understand the damage caused to the child by sharing the re-reading of the individual and the relational meanings of prejudicial behaviors; (d) the ability to take on one’s responsibilities and activate restorative behaviors as a function of change; and (e) an initial ability to participate in a restorative intervention project.

Step VII: The last phase of the clinical work involved the supervision of the treatment and parenting care interventions that were initiated by the psychologists from the various teams with the parents who were assessed as therapeutically treatable.

It is necessary to consider a multi-method, multi-approach, multi-orientation clinical work setting for the treatment of complex trauma. This is integrated with the idea that it is necessary to have therapists with differing knowledge and skills, who must be able to rely on a very rich and articulated toolbox to help parents process their traumatic experiences and find new, more adaptive strategies to cope with external reality and compromised parenting relationships. From this perspective, EMDR represents, as recognized by the National Institute of Mental Health, an effective therapy for the re-processing of trauma, and offers an innovative way to both understand the psychopathological outcomes and to carry out a specialist intervention, regardless of the reference approach. In particular, the supervision work focused first of all on the definition of a precise and articulated conceptualization of the case and of the consequent therapeutic plan for each single parent. Considering the complexity and the pluri-traumatization of the parents involved, it would have been unthinkable (even considering the project times) to deal in a chronological and consequential way with all the traumatic experiences lived. Therefore, we chose to undertake the clinical work with the highest disturbance and impact on the current exercise of parenting. The protocol envisaged the use of (target) memories from childhood experiences, in particular, with reference to the items of the Adult Attachment Interview (AAI), which refer to the experiences of relationships with parents before the age of 12 (significant memories reported in support of the chosen adjectives). In addition, the float back technique was used, and we proposed the use of current stories of caring relationships that put parents in difficulty by inducing traumatic experiences, in a bid to go back to when the parent had experienced those negative emotions, sensations or self-beliefs for the first time.

Through the supervision work, it was possible to record the strong motivation of the parents involved to expose themselves in this therapeutic process, which appeared to be in line with an intrinsic and spontaneous request for help for themselves and their children. The parents, once they understood the intergenerational harmful effects of the transmission of the unprocessed trauma, appeared tense and willing to change its inevitable course, feeling the fatigue and suffering associated with a re-contact with memories that had apparently fallen into oblivion. However, they also felt the relief and the comfort of being able to free oneself and one’s children from the shackles of diminished and compromised life prospects.

## 5. Discussion

After the qualitative and observational analysis of the outcomes of the interventions implemented in this project, we note some differences between our sample and the recent national survey, promoted by the Guarantor Authority for Childhood and Adolescence and carried out by Terre des Hommes and CISMAI [32,33], in collaboration with ANCI (italian acronym of National Association of Italian Municipalities), in which the prevalence of ACEs was higher in female minors. On the other hand, there is a similarity with the data of the metropolitan area of Naples (Italy) in which, after a period when the genders were equally present, there was an imbalance in favor of males.

With respect to the age of the minors, the number of subjects in the pre-adolescent and adolescent range appears to be more frequent, as can also be seen in the national data. This data is strongly connected to the increase in the self-determination capacity of children who are able to express their discomfort over the years, as well as to the greater presence of reference adults, such as teachers and operators of the territories, who can more easily recognize the malaise of a child and activate the network. Another element that marks out this data is the observation that the specialized interventions proposed within the project come at the end of a previously fragmented path of multiple interventions that have alternated over the years.

The phenomenon of assisted violence emerged in the four territories of the Campania region, and is reflected in this study, appearing to be a consistent trend. In recent years, an interesting awareness campaign on the phenomenon of gender-based violence has been launched on the national and international scene, which, on one hand, has allowed women to have a greater awareness of their rights and the opportunities that derive from liberation from situations of severe violence. On the other hand, it has allowed network operators to increase their expertise on the subject, with regard to the rights and needs of female victims of violence and, above all, in relation to witnessed violence.

The other relevant data are related to maltreatment and neglect. These often emerge in connection with other phenomena, such as sexual abuse and witnessed violence.

The small number of cases of sexual abuse that have emerged confirm the national data, which show that the phenomenon, also underestimated in the national research, is affected by difficulties with detection. This is attributable, in the first place, to the emotional resistance already present with respect to any form of maltreatment on the subject. In childhood, this is more consistent with respect to sexual abuse and, moreover, the concerns, and sometimes a paucity of skills, regarding protection and the assistance paths to be activated. This is more evident in territorial areas where there are no specific multidisciplinary team services dedicated to supporting the detection and protection processes.

With regard to children in situations of witnessed violence, an important reflection has been made regarding how to manage situations of domestic violence, especially with regard to the issue of parenthood.

As can be seen from the data, the number of violent situations witnessed by children was very consistent, as reported by social services, communities and anti-violence centers.

The need to diversify the intervention with parents, who are not in the same position with respect to the children, has thus been problematized. Therefore, it is necessary to differentiate between the path of the victimized mother, who, by suffering male violence, inevitably also weakens the parental function, and the father, who is the perpetrator of direct violence against the partner and at least indirectly against the children.

Obviously, the situations that have arisen are also extremely varied in a way that mirrors larger case histories: there are mothers who asked for help for themselves and for their children, even in a welcome path; children who were placed in protection without their mothers because male violence was not initially detected, but family relationships were interpreted in terms of discomfort or conflict; mothers who, over time, left the protective contexts without their children; and mothers who left the protection together with their children.

The project made it possible to discuss the resistance of the territory to recognize the impact of violence on parenthood and the tendency, in a prejudicial way, to split the paths between the protection of the children and help for the mother. This was done without considering the seriousness of the impact of the separation on children, who often, in these situations, experience a deep alarm for what is happening to their mother.

It is possible to further reflect on the modalities of access to taking charge in the experimentation.

In the network that supported access and ongoing participation in the project, it was possible to detect, in addition to the territorial social services, a substantial share of residential facilities for children, and mothers and children, or centers for women victims of maltreatment with children.

Despite the presence of so much wealth, we have often found ourselves in unclear or undefined intervention contexts between consensuality and protection. In some areas, there are inconsistent measures limiting parental responsibility, and therefore, there are no provisions relating to the evaluation of parenthood. Sometimes, a “strengthening of parenting skills” is recommended, but this is not feasible without a preliminary assessment of recoverability and the associated positive prognosis.

With regard to spontaneous access, another recurring situation occurred, namely there was a presence of a partially protective parent, an absence of measures limiting parental responsibility and no requirements to engage in assessment and support processes. Therefore, there have often been “false starts” due to participants evading the situation, leading to the suspension of the interventions.

The frequent absence of an explicit judicial mandate on evaluation in such indefinite intervention contexts requires necessary and demanding preliminary work, either with the parents to build, where possible, an authentic consent, or with the Juvenile Court to receive an explicit mandate.

Thanks to the interventions carried out by the teams on these cases, it has been possible to change the chronic situations with regard to the suffering of the minors sent to the project by supporting de-institutionalization processes and building alternative life projects in agreement with the AG Judicial Authorities and the local services. This has offered the young victims and their families a real opportunity for change, and a chance to restore the conditions of well-being for the growth of the children.

## 6. Conclusions

The intervention and treatment strategies with regard to ACEs emerged after a deep reflection on the processes activated within the project, which took into account the contextual elements, the organizational structures, the actions of psychosocial management and clinical intervention, and the constraints and opportunities, in order to systematize the transformative opportunities achieved through the experiment [34,35].

In the context of the interventions with children exposed to ACEs, the clinical supervision work has a double relevance. On one hand, the delicacy and complexity of the clinical work in such extreme situations regarding the treatment of complex trauma, in which one is impacted by stories of pain and despair that are often unthinkable and unspeakable, has also given space and relevance to the process of listening and sharing the countertransference dimensions that are inevitably activated within the clinical path. This has triggered a reflective and elaborative reaction. In this sense, the group has assumed the functions of holding, comforting, consolation and protection, allowing all participants to leave their therapy rooms, where often, in solitude, they were deeply affected by experiences of impotence, resignation, professional inadequacy, failure, pain, horror, anguish, disgust, anger, rejection, loss and confusion, in order to share their experiences and find space for both emotional and professional reflection on their usual operational practices. As Hermann [36] states in her book *Trauma and Recovery*, trauma is contagious and traumatic countertransference reactions are inevitable. From this point of view, the containment function of traumatic pain (a reflection of the victim’s experience) makes contact with such highly traumatic experiences conceivable, preventing the risk that, in the clinical operator (even the most experienced clinicians), defenses of avoidance and feelings of helplessness are activated. There is despair in the face of a child’s pain and desire for adequate care. The same avoidance defenses can be activated against the parent (the executioner), supported by a jumble of confused emotions, such as anger, disgust, helplessness, guilt and shame. These countertransference reactions inhibit professional skills and risk causing the taking charge process to fail. Emotional, reflective supervision uses group time and space to reflect on their own experiences, exploring feelings about work without fear of being judged by supervisors or groups of colleagues. This supportive role of the group, supported by an experienced supervisor, presupposes that the initial ideas have been developed in the group and that a cooperative working climate has been created where therapists belonging to different services also allow themselves to share vulnerable aspects of their professional experience.

The other aspect of supervision concerns the continuous improvement of the methodologies. Specifically, for some MST professionals, this involves the use of diagnostic tools and approaches in therapy. This task of supervision in the work on maltreatment is of great importance since, on one hand, the unthinkability of the issues supports the risk of prolonging the time of diagnosis, which increases the risk of remaining in diagnostic doubt. On the other hand, the burden intertwining with judicial processes often exposes the clinician to having to support the outcome of the path (of the psychodiagnostic evaluation of the child and the recoverability of the parent), with a heavy responsibility towards the child and towards the possibility of seeing the ties with the child’s parents being broken. These critical nodes often sustain a dangerous impasse in the handling paths, with serious consequences. Consider, for example, the long stays of children in shelters without the doubts about the conditions that led to the expulsion being eased. In such circumstances, it is not possible to undertake a path of repair or a replacement of ties. Supervision, from this point of view, reinforces the experience of professional competence and efficacy, making it possible for the clinician to expose himself, through his own professional judgment, to the judicial context.

The supervision meetings on child care focused on psychodiagnosis as an analysis of the prevailing post-traumatic functioning, not only in terms of descriptive diagnosis, and, compared to the clinical intervention, much space was also dedicated to EMDR supervision. Overall, we emphasized how crucial it is for the clinician to consider that the articulated intervention in these situations requires the ability to work simultaneously to allow for a change in the child’s distorted perception of themself, the world and relationships; while at the same time, to try to act on the external world, allowing the child the possibility of having a reparative experience, which, by contrasting it with the view of the world as malevolent and threatening, can lay the foundations to make it possible to change the distorted system of meanings.

The supervision of the evaluation of parental recoverability, in particular, worked on the possibility of comparing it with the experimentation of a clinical intervention model that conceptualizes the evaluation as a procedural diagnosis, and does not refer to it as a nosographic label.

## 7. Limitations

The present study revealed both a qualitative and quantitative nature, in reference to the structured methodological approach. In the first analysis, our sample of subjects was limited in size, as both the minors and the parents who had been taken into care had come from reports made by the legal institutions of territorial jurisdiction that had tried to reconcile specific paths of protection of the minor, both from a legal point of view and from a psychological support perspective. Furthermore, often only the minor victims of ACEs are cared for, or only the parental couple (which is not always made up of both parents). In the second analysis, it would be advisable to define and structure the work of a multidisciplinary team in the best possible way according to a specific protocol, so as to be able to use the same methodology in different contexts to verify and monitor further environmental and socio-cultural variables.

## Data Availability

Written informed consent was obtained from the subject(s) in order to publish this paper.

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
