# Peer review of "The Multilevel Pathway in MSTs for the Evaluation and Treatment of Parents and Minor Victims of ACEs: Qualitative Analysis of the Intervention Protocol"

_children, 2022, doi:10.3390/children9030358_

Round 1

Reviewer 1 Report

I have read the article several times trying to understand the research but I have huge doubts

  1. the sample is 35 subjects (1/3 minors and 2/3 parents?)? I feel that such a sample does not allow any conclusions
  2. even considering a qualitative study, the results  of the intervention are not ver clear
  3. I am not sure that the experience is reproductible by others

Author Response

Reviewer 1

I have read the article several times trying to understand the research but I have huge doubts

  • the sample is 35 subjects (1/3 minors and 2/3 parents?)? I feel that such a sample does not allow any conclusions
  • even considering a qualitative study, the results  of the intervention are not ver clear
  • I am not sure that the experience is reproductible by others

Authors: The Authors thank the Reviewer for taking the time to read our paper. For the valuable suggestions indicated, we have clarified many of the aspects detected by inserting the paragraph "Limitations".

Reviewer 2 Report

Preliminary note: As announced before taking over the review, I am not a psychologist, but assess the paper from the perspective of qualitative research. This may explain why my review primarily focuses on methodological issues.

Review:

The described intervention seems to me to be an important project that touches many relevant aspects, shows the multifactorial causes for Adverse Childhood Experiences and promises to be able to offer differentiated support for the affected children and their families through the multi-layered approach. In addition, the approach via multidisciplinary specialized teams seems very convincing to me in this situation.

However, in methodological-methodological terms, important questions remain unresolved in the current form of the manuscript: Who was involved in the intervention and how? How exactly were observations made and interviews (with which questions) conducted by whom? Who evaluated the observations and interviews and how exactly? How were relevant categories such as "denial of facts", "denial of awareness", "denial of responsibility" and "denial of impact" developed? Was the approach always primarily deductive, or was it also inductive?... In this regard, the paper would need to be revised to clarify these issues.

And it seems to me that the question in which form the researchers were involved in the analysis of the object of the analysis is very elementary. In my opinion, this should definitely be (self-)critically reflected upon. In the abundance of facets of the interventions and data collection within the project described, it seems difficult to present all of this. Therefore, I would recommend to focus on selected aspects in the article.

Author Response

Reviewer 2

The described intervention seems to me to be an important project that touches many relevant aspects, shows the multifactorial causes for Adverse Childhood Experiences and promises to be able to offer differentiated support for the affected children and their families through the multi-layered approach. In addition, the approach via multidisciplinary specialized teams seems very convincing to me in this situation.

Authors: The Authors would like to thank for the feedback and suggestions that will surely improve our ducks. We have marked the changes in red.

However, in methodological-methodological terms, important questions remain unresolved in the current form of the manuscript: Who was involved in the intervention and how? How exactly were observations made and interviews (with which questions) conducted by whom? Who evaluated the observations and interviews and how exactly? How were relevant categories such as "denial of facts", "denial of awareness", "denial of responsibility" and "denial of impact" developed? Was the approach always primarily deductive, or was it also inductive?... In this regard, the paper would need to be revised to clarify these issues.

Authors: In paragraphs 3.1, 3.2, 3.3 and 4.2 we have specified the operators involved in the various phases and coem the phases of assessment and clinical interview have been structured. Furthermore, we have specified the importance of the levels of "parental denial" in order to further clarify the importance of detecting these levels in order to better structure the support and therapeutic path for the protection of the minor.

And it seems to me that the question in which form the researchers were involved in the analysis of the object of the analysis is very elementary. In my opinion, this should definitely be (self-)critically reflected upon. In the abundance of facets of the interventions and data collection within the project described, it seems difficult to present all of this. Therefore, I would recommend to focus on selected aspects in the article.

Authors: We appreciate the input of this suggestion and have reflected on the importance of presenting such methodological work to highlight the work of a multidisciplinary team. Furthermore, in order to evaluate the effectiveness of our therapeutic and care pathway, we felt the need to insert all the steps. However, we have also implemented this aspect in the "Limitations" section.

Round 2

Reviewer 2 Report

Even though some methodological aspects are not addressed in depth in the present form of the text, one now gains an adequate impression of this important project.